# Adeno-associated virus capsid assembly is divergent and stochastic

Tobias P. Wörner[1,2], Antonette Bennett[3], Sana Habka[1,2], Joost Snijder[1,2], Olga Friese[4], Thomas Powers[4], Mavis Agbandje-McKenna[3] & Albert J. R. Heck [1,2✉]

Adeno-associated viruses (AAVs) are increasingly used as gene therapy vectors. AAVs package their genome in a non-enveloped $T = 1$ icosahedral capsid of ~3.8 megaDalton, consisting of 60 subunits of 3 distinct viral proteins (VPs), which vary only in their N-terminus. While all three VPs play a role in cell-entry and transduction, their precise stoichiometry and structural organization in the capsid has remained elusive. Here we investigate the composition of several AAV serotypes by high-resolution native mass spectrometry. Our data reveal that the capsids assemble stochastically, leading to a highly heterogeneous population of capsids of variable composition, whereby even the single-most abundant VP stoichiometry represents only a small percentage of the total AAV population. We estimate that virtually every AAV capsid in a particular preparation has a unique composition. The systematic scoring of the simulations against experimental native MS data offers a sensitive new method to characterize these therapeutically important heterogeneous capsids.

[1] Biomolecular Mass Spectrometry and Proteomics, Bijvoet Center for Biomolecular Research and Utrecht Institute for Pharmaceutical Sciences, University of Utrecht, Padualaan 8, Utrecht, The Netherlands. [2] Netherlands Proteomics Center, Padualaan 8, Utrecht, The Netherlands. [3] Department of Biochemistry and Molecular Biology, Center for Structural Biology, the McKnight Brain Institute, 1200 Newell Drive, Gainesville, FL, USA. [4] Biotherapeutics Pharmaceutical Sciences, Pfizer WRDM, 700 Chesterfield Parkway W, Gainesville, MO, USA. ✉email: A.J.R.Heck@uu.nl

Adeno-associated viruses (AAVs) are small, non-pathogenic, ssDNA packaging viruses, capable of infecting a wide range of vertebrate hosts, including humans. AAVs belong to the *Parvovirinae* subfamily of the *Parvoviridae*, and *Dependoparvovirus* genus. As the name implies, they require co-infection with adeno- or herpesviruses as helpers for replication[1–5]. AAVs package a 4.7 kb genome encoding non-structural (*rep*), structural (*cap*), assembly activating (*aap*), and membrane associated accessory (*maap*) proteins[2,6–8]. AAVs have become widely used for gene therapy applications, with several advantages over other viral vectors, including a lower toxicity and the availability of over 150 naturally occurring genotypes and serotypes[9–11]. These serotypes differ in their tropism, and thus can target most tissues and cell types for gene delivery[12]. Recombinant AAVs (rAAVs), packaging a gene of interest (GOI), have been successfully studied in clinical trials for the treatment of a wide variety of rare genetic disorders. Notably, three AAV-based biologics have been approved: Luxturna by the FDA and EMA (FDA STN#125610; EMEA/H/C/004451), Zolgensma by the FDA (FDA STN #125694), and Glybera by the EMA;[13–16] and several other products are presently being reviewed (https://clinicaltrials.gov/)[17]. In these applications, the GOI replaces the natural AAV genome for delivery to tissues or cells to treat a monogenic disease. In addition, rAAVs are widely used research tools for transgene expression in tissue culture and preclinical animal models[18].

AAV capsids consist of a total of 60 molecules of viral proteins (VPs); a mixture of the three overlapping gene products, VP1, VP2, and VP3, encoded by the *cap* open reading frame (ORF) and organized in $T = 1$ icosahedral symmetry (Fig. 1)[19]. The VPs are generated through alternative splicing of the mRNA and use of an alternate translational start codon[20]. The VP3 (59–61 kDa, 524–544aa) sequence is shared among all VPs and is referred to as the VP3 common region. VP2 (64–67 kDa, 580–601aa) is approximately 57aa longer than VP3 and the VP2 N-terminal region is referred to as the VP1/VP2 common region. VP1 (79–82 kDa, 713–738aa) is approximately 137 aa longer than VP2 and this region is called the VP1 unique (VP1u) region. The VP3 common region assembles the icosahedral capsid. The VP1u contains an essential phospholipase A2 (PLA2) enzyme, and VP1u and VP1/VP2 common region contain nuclear localization sequences (NLSs)[21]. These N-terminal extensions of VP1 and VP2 are reported to play crucial roles in endosomal trafficking and escape, nuclear localization, and genome release (reviewed in ref. [22]). Specific VP subunits have been targeted for modification, such as the removal of common immunogenic motifs[23], the integration of sequences encoding for fluorophores[24] or designed nanobodies against receptors on target cells, redirecting tropism[25]. As some of these modifications are directed to VP1, and because the unique portions of VP1 and VP1/VP2 are required for infection, it is important to assess how many VP1 and VP2 molecules are in the AAV icosahedral capsid and how they are, if at all, structurally organized.

The AAV capsid composition of VP1:VP2:VP3 is estimated to be in a ratio of 1:1:10 based on gel densitometry and mass spectrometry studies[2,26–28]. The atomic resolution structures of several primate AAVs, and recently the non-primate BatAAV, have been determined by cryo-electron microscopy (cryo-EM) with image reconstruction and X-ray crystallography. In all high-resolution structures, only the overlapping VP3 region is resolved, VP1u, VP1/VP2 common region, and the first ~20 amino acids of VP3 were not observed (reviewed in ref. [29]). This is likely due to the low copy number of VP1 and VP2 or this region may have high flexibility due to the glycine rich sequence in the VP1/VP2 common region (reviewed in ref. [29]). This was confirmed by disorder prediction of the VP1 sequence of several AAVs[30]. This

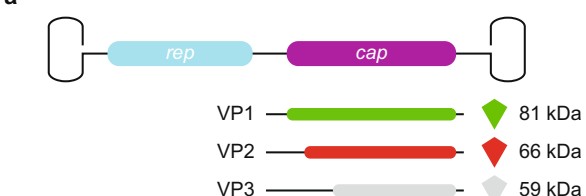

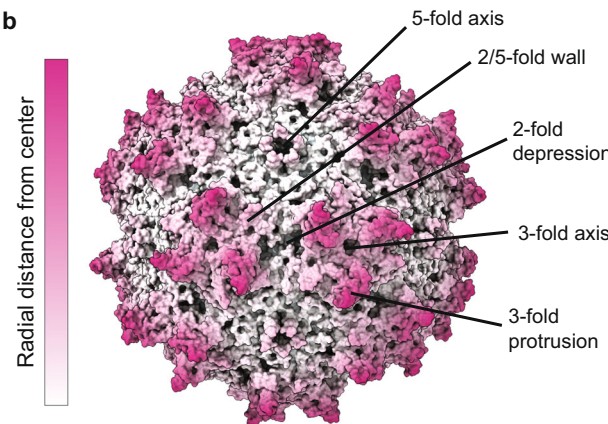

**Fig. 1 Adeno-associated virus (AAV) capsid structure. a** The capsid is composed of 60 copies of a combination of VP1, VP2, and VP3. For the VPs we use the green, red, and gray color scheme, respectively, throughout the manuscript. The three VPs are encoded within the same cap ORF where the different VPs are produced via differential splicing and alternate start codons. All VPs share a common C-terminal VP3 sequence. **b** An AAV1 structure (PDB ID: 3NG9) radially colored from the capsid center (white to purple). The approximate icosahedral 2-, 3-, 5-fold axes along with the 2/5-fold wall are labeled. The icosahedral symmetry is retained by VP3, whereas the VP1 and VP2 extensions, and the variable stoichiometry result in a breakdown of this symmetry.

high flexibility of the VP1/VP2 common region is not amenable to the icosahedral symmetry imposed during structure determination. Unresolved protein globules have been observed on the capsid interior of low resolution cryo-EM density maps of AAV1, AAV2, and AAV4, and they have been predicted to be the VP1u, and/or the N-terminal of VP2[31–33]. Consistent with a WT infection, in rAAV production the different VPs are expressed in a 1:1:10 ratio, which leads to a widely assumed average capsid stoichiometry of 5:5:50 (#VP1:#VP2:#VP3). More recently, it has been discovered that the VP ratio is dependent on the production system, with VP1 having generally lower expression levels in baculovirus production systems[34,35]. The available structures or capsid stoichiometry do not answer the questions whether the capsids assemble with a defined or variable ratio of VPs, and how the lower abundant VP1 and VP2 subunits are organized within the icosahedral capsid. Preliminary native mass spectrometry data suggested a stochastic incorporation of VP1, VP2, and VP3 hinting at the co-occurrence of AAV capsid particles of highly variable compositions[28,36]. However, considering the importance of the unique role of the VP1u and VP1/VP2 common region, each AAV particle should contain at least one copy of each VP to properly function. Production of AAVs is expensive and administration in high doses can cause an adverse immune response, problematic for applications in gene therapy. Hence, a more detailed understanding of AAV capsid assembly, connecting bulk VP1, VP2, and VP3 expression to the assembled capsid stoichiometries, is imperative for the development of AAV particles, useable in gene therapy.

**Table 1 Overview of analyzed AAV samples.**

| Serotype | Name | Expression system | Stoichiometry by LC-UV/MS (VP1%:VP2%:VP3%) | Best scoring simulation (VP1%:VP2%:VP3%) |
|---|---|---|---|---|
| AAV1 | AAV1 VP3-only | SF9 | — | 0:0:100 |
| AAV1 | AAV1 | SF9 | — | 4:1:95 |
| AAV8 | AAV8_1 | SF9 | — | 1:3:96 |
| AAV8 | AAV8_2 | SF9 | 1:9:90 | 1:10:89 |
| AAV5 | AAV5 | SF9 | 1:4:95 | 1:5:94 |
| AAV9 | AAV9 | HEK293 | 6:16:78 | 6:14:80 |

Overview of analyzed AAV serotypes with their origin, reference name used in this studies, and expression platform alongside the experimentally determined bulk VP ratios (if available). The best matching bulk VP ratio is based on simulation scorings shown in Fig. 5.

Here, by using an Orbitrap UHMR, with improved capacity to efficiently transfer and detect very high mass ions[37], we perform an in-depth native mass spectrometry analysis of AAV particles from different serotypes and from different production platforms. The improved sensitivity allowed us to record well-resolved mass spectra. We performed theoretical simulations that model complicated mass spectra, being hampered in certain $m/z$ windows by strong interferences, whereas in other areas they become well-resolved. The simulations helped us understand these spectral features and to predict which part of the spectra are most informative. Based on the native mass spectra and the concomitant simulations we demonstrate that all studied AAV serotypes consist of heterogeneous populations with variable VP stoichiometries. The data confirm an assembly model in which AAV particles assemble by 60 random draws from a mixed pool of VP1, VP2, and VP3 in a ratio determined by their relative expression levels. Scoring the native MS data against simulated spectra based on the stochastic model provides a sensitive and accurate estimate of the VP ratios in the capsid. This random assembly breaks symmetry in the AAV capsid beyond the VP3 common region, and may thus explain why the unique functional regions, e.g., VP1u and the VP1/2 common region, of the AAV capsid are not readily observed in other structural biology approaches.

## Results

**AAV capsid heterogeneity.** Here, we investigated AAVs of different serotypes, produced in different facilities, and using HEK293 cells or the baculovirus/SF9 system (see Table 1) and evaluated whether high-resolution native MS analysis of the empty capsids using the Orbitrap UHMR platform with enhanced transmission at the extended mass range[37], would enable us to determine accurately the mass and exact composition of each of these gene delivery platforms. Our selection includes two varieties of the AAV1 serotype: complete capsids assembled from all three VPs, as well as VP3-only capsids that lack both VP1 and VP2. As illustrated by SDS-PAGE and negative-stain electron microscopy of the samples, both variety of AAV1 assemble into icosahedral particles, and while the complete capsids contain a VP1:VP2:VP3 ratio close to the expected 1:1:10, the VP3-only capsids indeed show only a single band for VP3 on SDS-PAGE gel (Fig. 2a). The pair of AAV1 samples thus serve as useful controls to evaluate how the mass distribution of AAV capsids relates specifically to VP composition.

As expected, the VP3-only AAV1 yielded well-resolved native mass spectra containing a single charge state distribution around 21,000 $m/z$ with a calculated mass of $3,571 \pm 0.3$ kDa (Fig. 2b). This experimental mass deviates by only +2.8 kDa (i.e., 0.08%) from the expected mass of a $T = 1$ icosahedral capsid composed of 60 VP3 subunits, likely due to remaining solvent adducts. To aid in the interpretation of the native mass spectra, a Python class

was developed to simulate the native MS spectra enabling a direct comparison with the experimental data (Fig. 2b, c). The simulations mimic the $m/z$-dependent resolution for each ion species at a given transient time in the Orbitrap (see method section and Supplementary Fig 1 for a detailed explanation). The simulated mass spectrum for this sample closely matches the experimental spectrum, with peak centroids deviating only by 3.1 Th on average. This result confirms that AAV capsids assemble with a high fidelity into the $T = 1$ icosahedral capsids with exactly 60 subunits, without any substantial defects or misassembled structures being present.

In contrast to the homogeneous VP3-only AAV1 capsids, wild-type AAV capsids are composed of VP1, VP2, and VP3. Analyzing such wild-type capsids yields much more complex native mass spectra (Fig. 2c). At first glance, this native mass spectrum seems to be composed of three partly resolved charge state distributions, with successive mass differences of around 6.5 kDa. Following a conventional charge state assignment strategy, the assigned masses correspond to a stoichiometry of 2:2:56 (#VP1:#VP2:#VP3), with the successive +6.5 kDa mass differences attributed to additional VP3-to-VP2 substitutions. However, overlaying the experimental data with simulated spectra for a range of different VP stoichiometries reveals that ion signals originating from AAV1 capsids of different composition overlap substantially, and are hard to resolve in the $m/z$ dimension. The simulations reveal that ion signals from AAV capsids of different stoichiometry can substantially interfere with each other, complicating the appearance (and interpretation) of the native mass spectra. In particular, the mass difference of three subsequent VP3-to-VP2 substitutions coincides precisely with the next charge state in the original series of peaks of the capsid without those substitutions. Similarly, the mass difference of three VP3-to-VP2 substitutions is similar to that of a single VP3-to-VP1 substitution and produces coinciding peaks in the native MS spectra. These simulations therefore reveal that the complexity of the AAV capsids may extend much further than the apparent three series of charge states that are resolved in the mass spectrum (Fig. 2c and Supplementary Fig 1). Notably, this means that our earlier report on the analysis of the composition of AAV capsids, likely was somewhat ambiguous[28]. These new UHMR-based analyses reveal definitively that wild-type AAV1 capsids do not assemble into a single well-defined VP stoichiometry, but rather consist of a heterogeneous mixture with varying compositions.

**Stochastic capsid assembly.** Based on these observations, we propose a stochastic model of AAV capsid assembly, in which particles assemble by 60 random draws from a mixed pool of VP1, VP2, and VP3, without any organizing principle to determine the final VP stoichiometry, other than the relative expression levels as depicted in Fig. 3a. One additional contributing

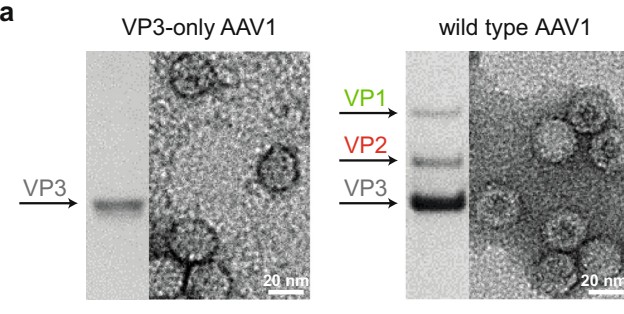

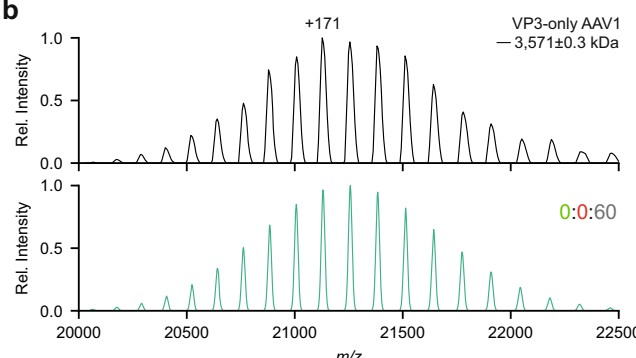

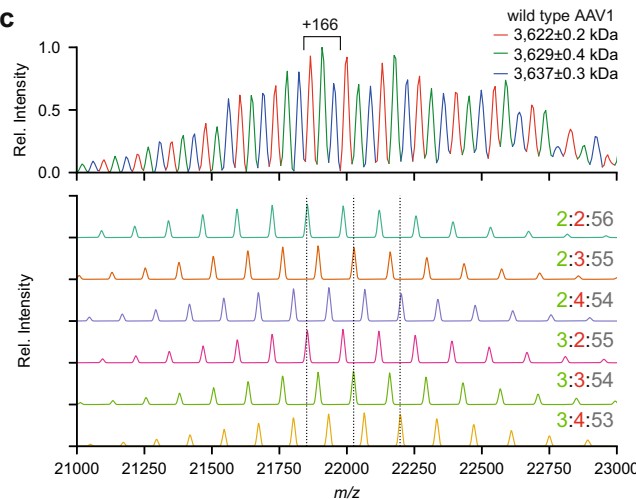

**Fig. 2 Adeno-associated virus (AAV) capsid structure and heterogeneity.**
**a** Coomassie stained SDS-PAGE and negative stain EM images of VP3-only AAV1 (left) and wild-type AAV1 (right) capsids. **b** Native MS spectrum of the VP3-only AAV1 capsids, with an assigned mass of 3,571 ± 0.3 kDa (masses are reported as mean ± s.d.), is shown at the top. The simulated native spectrum for capsids with a 0:0:60 (#VP1:#VP2:#VP3) stoichiometry is shown below and accounts for all observed peaks in the experimental data. **c** Native MS spectrum of wild-type AAV1 capsids, showing three peak series with a relative mass shift of around 6.5 kDa. The simulation of several capsid stoichiometries are shown below and illustrate the potential coinciding charge state distributions caused by VP3-to-VP2 and VP3-to-VP1 substitutions. VP ratios are indicated for each simulation respectively (#VP1:#VP2:#VP3).

factor is that AAV capsid assembly is known to utilize the assembly-activating protein AAP, which is required for VP oligomerization, stabilization, and transport to the nucleus where the process of capsid formation occurs[7,38]. Whether AAP preferentially incorporates some VPs over others is currently not known, but such preference would shift the 'relative VP

expression levels' that we refer to in our stochastic assembly model throughout this report.

According to our model there is a theoretical total of 1,891 possible co-occurring capsid stoichiometries (based on $n = 3$ different VPs with $k = 60$ subunits total, giving $(k + n-1)!/(k!*(n-1)!)$ unique combinations/masses). The probability $P$ of a given stoichiometry is given by a multinomial distribution: $P$(VP1, VP2, VP3) = 60!/(VP1!VP2!VP3!) * $p_{VP1}$^VP1 * $p_{VP2}$^VP2 * $p_{VP3}$^VP3, where VP1, VP2, VP3 denote the number of the respective VPs, $p_{VP}$ is the probability of drawing a given VP, and $p_{VP1} + p_{VP2} + p_{VP3} = 1$. Depicting the probability for each combination of VP3-to-VP2 and VP3-to-VP1 substitutions in heat-maps then provides a useful overview of predicted AAV capsid compositions.

This is shown in Fig. 3b for a 3:10:47 bulk average ratio of VP1:VP2:VP3, as experimentally determined for AAV9 capsids by LC–UV/MS (see below). In the stochastic assembly model, even the single-most abundant VP stoichiometry represents a mere 3% of all capsids, highlighting how heterogeneous the total population of AAV capsids can be. As shown in Fig. 3c, the majority of the capsids contain between 0–10 copies of VP1, 2–20 copies of VP2, and between 35–55 copies of VP3.

**AAV spectra simulation and scoring.** The exact distribution of capsid masses can also be predicted by our stochastic assembly model, by calculating the theoretical mass of the 1,891 possible VP stoichiometries and plotting them against their estimated probability (Fig. 4a). This distribution spans more than 300 kDa of very densely populated masses between 3.6 and 3.9 MDa. A similar approach has previously been reported to predict the extent of the mass distributions for AAV capsids analyzed by charge detection mass spectroscopy (CDMS)[36]. For comparison to our experimental data, this mass distribution is converted to the mass-to-charge dimension, assuming normal (Gaussian) charge state distributions. At infinite mass resolution the stochastic assembly model predicts a very densely populated $m/z$ spectrum, which when simulated at successively lower mass resolving powers, gradually collapses into smaller series of resolvable peaks, until three apparent charge state series are visible, exactly as observed in our experimental AAV1 native mass spectra in Fig. 2c, and for AAV9 shown in Fig. 4b, c. These simulated spectra based on the stochastic assembly model, using all possible 1,891 AAV stoichiometries with probabilities determined by the average VP1:VP2:VP3 ratio of 3:10:47 (as determined from the LC–UV/MS data on the monomers) closely matches our experimental data, supporting that the model indeed provides an accurate description of the AAV capsid assembly.

The very high number of theoretical ion signals at infinite mass resolution, originating from AAV capsids with different stoichiometries, thus collapse to a substantially lower number of distinguishable ion signals at (experimentally achievable) mass resolution (see Fig. 4d). Whereas it appears as though only three simple charge state distributions are present in our experimental AAV1 mass spectrum (Fig. 2c), these signals are unresolved composites of many unique ions, in which the relative contribution of each component determines the fine structure of the spectrum, such as the peak width, shape, and the precise position. By scoring the experimental spectra against the simulated spectra of the stochastic assembly model with systematically varied VP ratios, we found that the native MS measurements are in fact a very sensitive and precise measure of the AAV capsid composition and subunit stoichiometries (see method section and Supplementary Fig 2). Heat maps of this score, along with the frequency distribution of the VPs for the best-scoring ratio and a comparison between the

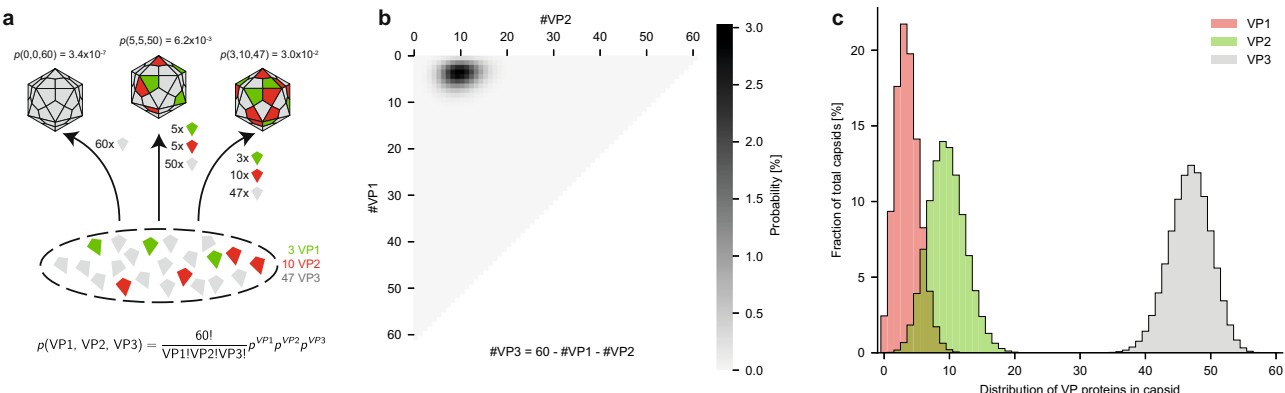

**Fig. 3 Stochastic assembly model for complete AAV capsids. a** Illustration of AAV capsid assembly following a stochastic model, where VPs are drawn randomly from the pool of expressed VPs. The probability for each capsid stoichiometry is solely determined by the ratio of the expressed VPs. Here displayed for AAV9 and the experimentally determined expression levels of 6% VP1, 16% VP2, and 78% VP3 or approximately 3:10:47 (VP1:VP2:VP3). The probabilities for exemplary VP stoichiometries are indicated above each capsid, respectively. **b** The calculated probabilities for all possible 1891 capsid stoichiometries are displayed as a heat map. The number of VP1 and VP2 in the 60-subunit capsids are indicated on the plot axis. **c** Summed probabilities for capsids having a specific number of VP1, VP2, or VP3 in their capsid.

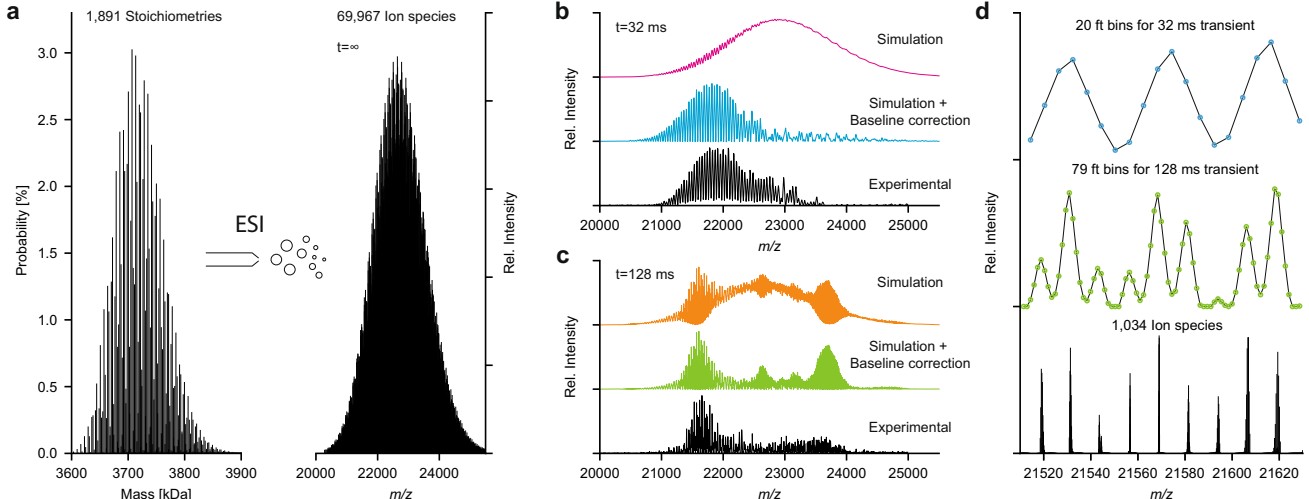

**Fig. 4 ESI and native MS of complex AAV samples. a** Theoretical mass distribution for all possible capsid stoichiometries of AAV9 and the resulting theoretical mass spectrum at infinite resolution. The mass distribution shows the 1,891 possible capsid masses with their probability as calculated in Fig. 2b. After ESI, each mass is simulated with 37 charge states yielding a total of 69,967 different ion species. **b, c** Simulated mass spectra for AAV9, showing the convoluted ion signals of the 69,967 different ion species shown in a) at resolutions, corresponding to transient times of 32 ms and 128 ms. Each plot shows (top to bottom) the simulated mass spectra before and after baseline correction with their experimental counterpart. Experimental data was recorded by transient averaging, thereby removing the unresolved regions presented in the non-baseline corrected. **d** Comparison (top to bottom) between 32 ms and 128 ms mass spectra and simulation at infinite resolution (data points/FT bins are shown as dots).

experimental and simulated spectra, are shown in Fig. 5 for several different empty AAV capsids from different serotypes and different production platforms; AAV1 VP3-only, AAV1, AAV5, AAV8, and AAV9. All tested serotypes show the same pattern in the native mass spectra, indicating that the stochastic assembly model applies broadly to all tested AAV serotypes. Notably, we also analyzed AAV8 capsids produced in the same production platform but produced in different laboratories. The shift in the VP ratios between these two preparations from the same host platform is robustly detected by our native mass spectrometry analyses. The determined bulk VP ratios for the two AAV8 preparations deviate only slightly by low percentage-points. However, these subtle shifts in the bulk VP ratios result in a twofold change in the number of capsids missing either VP1 or VP2, both important for infection and transgene delivery. Overall, the average VP ratios derived by native mass spectrometry, assuming the stochastic assembly

model, are in agreement with bulk VP ratios determined by LC–UV/MS.

For the AAV9 capsids, we were able to collect native mass spectra at even longer transient times of 128 ms. As shown in Fig. 6a at first glance these spectra look largely uninterpretable due to the extensive presence of interferences. However, based on the simulations, we can conclude that these interferences are not equally present in distinct m/z windows. In Fig. 6b, c we zoom in on such regions, which reveal even better resolved ion signals. In these parts of the spectra, as many as 11 distinct charge state series are resolved, confirming that the capsids are indeed highly heterogeneous, and confirming our stochastic assembly model. The experimental native mass spectrum is in very close agreement with the simulated spectra, further supporting that the model is an accurate description of the AAV capsid composition and that our simulations are essential for spectrum interpretation.

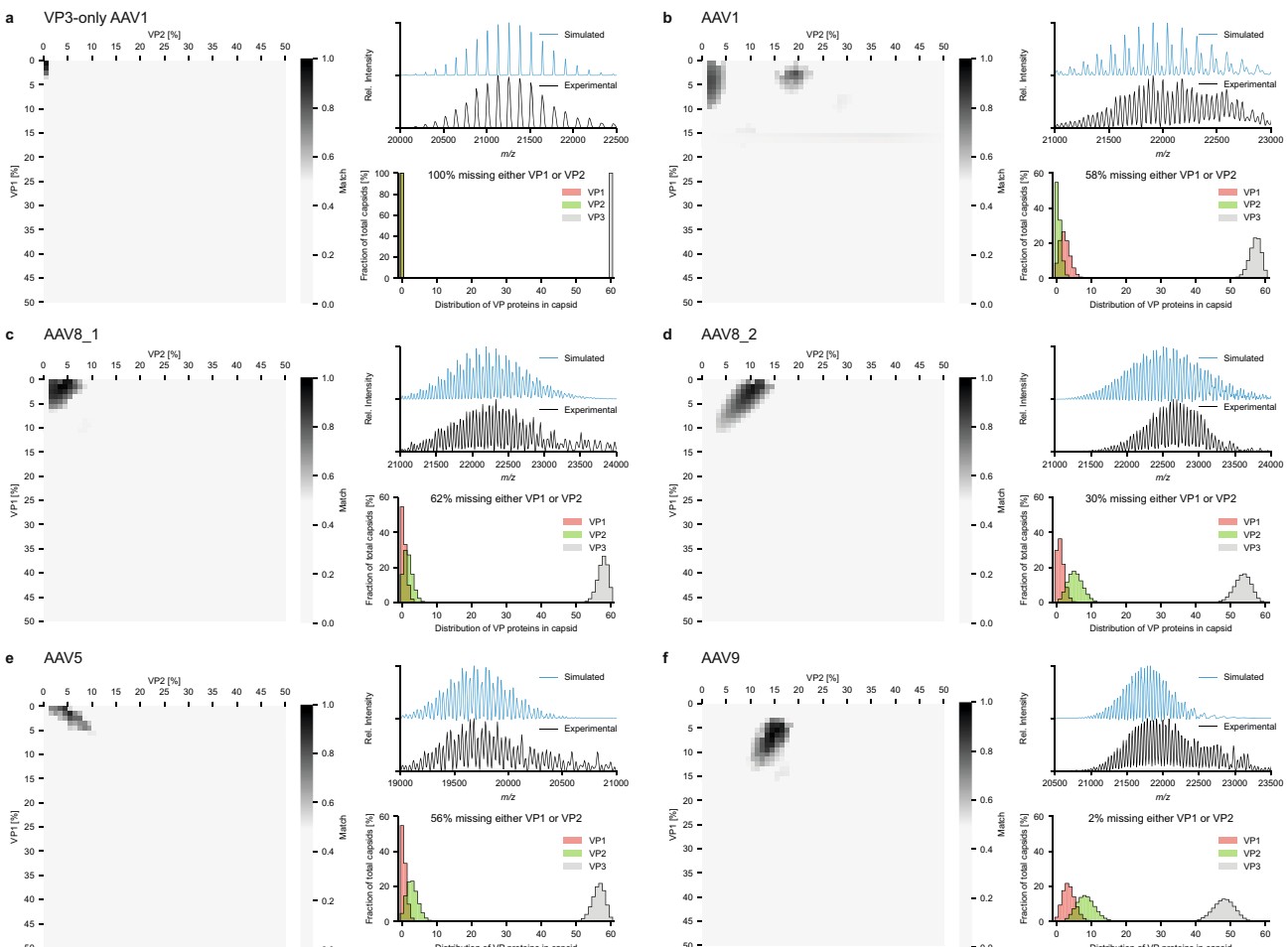

**Fig. 5 Top scoring optimized simulations for several distinct AAV products of different serotypes. a–f** Scores for the systematic comparison between simulations and experimental mass spectra are displayed for AAV1 VP3-only (**a**) and the wild-type capsids for AAV1 (**b**), AAV8_1 (**c**) and AAV8_2 (**d**), AAV5 (**e**), and AAV9 (**f**). AAV8_1 and AAV8_2 are different batches produced in different laboratories. Simulated mass spectra for the best scoring bulk VP ratios are shown with the experimental data alongside the corresponding probability histograms for capsids having a certain number of VP1, VP2, or VP3. Best scoring bulk VP ratios for each serotype are listed in Table 1. The amount of capsids missing either a VP1 or VP2 are indicated above the histograms and range between 100% for AAV1 VP3 only (**a**) and 2% for AAV9 (**f**).

## Discussion

**Random assembly**. Based on the high-resolution native mass spectrometry data and the parallel spectral simulations we demonstrate here that intact ~3.8 MDa AAV particles assemble by random incorporation of VP subunits. AAV are thus ensembles of widely divergent capsids with varying VP stoichiometries. For a given stoichiometry, we estimate that there are $60!/(VP1!VP2!VP3!)$ possible configurations with at best 60-fold redundancy due to the icosahedral symmetry of the particle, amounting to approximately $10^{12}$ unique capsid configurations for the widely assumed 5:5:50 ratio. Our model predicts that even the single-most abundant capsid composition represents less than 2.5% of the total capsid population. This suggest that the probability of finding a given AAV capsid with an exact composition and configuration of VPs is in the order of $10^{-14}$.

**Structural diversity**. This broad diversity of AAV capsid structures also explains why VP1 and VP2 remain elusive so far in structural studies by crystallography and cryo-EM, even at the nearly atomic resolution at which the shared VP3 sequence is described[31,39–42]. Whereas the common VP3 core assembles into an icosahedral structure, the stochastic composition and random incorporation of VP1 and VP2 makes the capsids decidedly

asymmetric and highly heterogeneous. As illustrated in Fig. 5, this heterogenous capsid population in a typical AAV prep may contain up to 60% particles completely lacking in either the VP1 or VP2 component for baculovirus derived capsids, or at least those with ratios well below the 1:1:10 expression level.

**Biological implications**. The evidence of divergent capsids with varying VP stoichiometries complicates the understanding of AAV transduction efficiency. VP1 and VP2 play a crucial role in endosomal trafficking, endosomal escape, nuclear trafficking, and genome release, and are thus essential components of rAAV during infection. Consistent with the role of the AAV VPs in the viral life cycle and its use in clinical gene therapy, VP1 and VP2 are not required for capsid assembly but the VP3 sequence alone is necessary to form the vector required to transport the therapeutic gene, although an increase in the expression of VP1 and VP2 has been described to generate a higher vector yield[43,44]. VP1 however, is important for transduction based on the presence of the NLS and PLA$_2$ domains and insufficient VP1 will lead to a reduced rate in vector transduction[34,35,45–47]. "Super-expression" of VP1 at a ratio of 1.9:0.1:8 (VP1:VP2:VP3, respectively) led to the production of morphologically similar capsids as those composed from standard triple plasmid infection

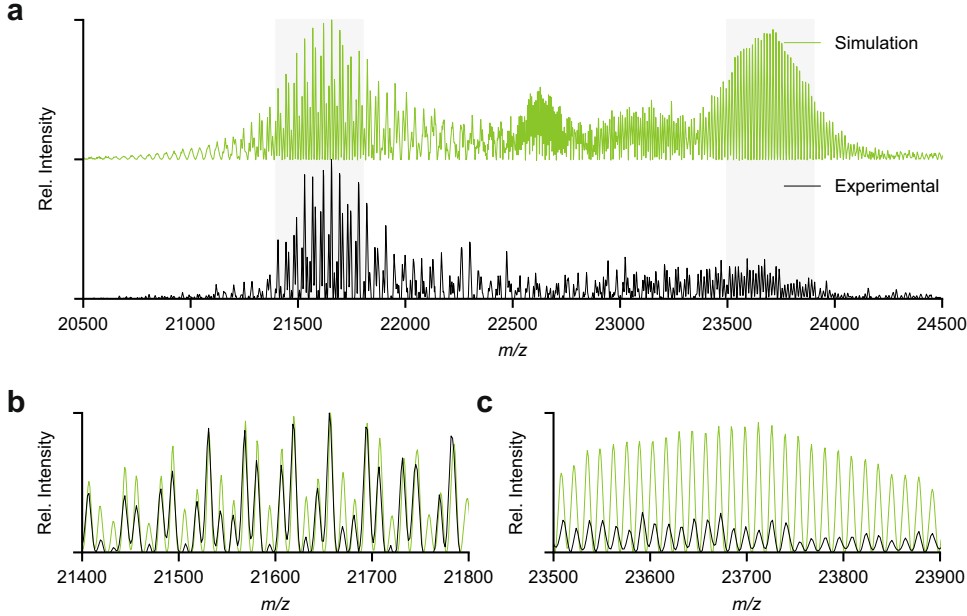

**Fig. 6 Resolved ion fine structure in the native mass spectra for wild-type AAV9. a** Baseline corrected simulation for AAV9, at a resolution corresponding to 128 ms transient time, with its experimental counterpart. **b**, **c** Overlay of resolved regions for the simulation with the experimental data.

methods. The super VP1 capsids also show higher transduction than its wild-type AAV counterpart[43]. Comparative analysis of the VP1 contents of the different AAVs as determined by native mass spectrometry show AAV9 has the highest VP1 content, followed by AAV1, and double the amount of VP1 produced in the AAV8_2 compared to AAV8_1 (summarized in Table 1). However, it is important to note that AAV9 was the only serotype manufactured by using a HEK expression system.

In summary, native mass spectrometry offers an exceptionally detailed picture of the diverse nature of AAVs, widely utilized for gene therapy applications. Although VP3 is the major capsid protein and accounts for the highest portion of the total number of VPs incorporated, the presence and abundance of VP1 and VP2 affect the biological efficacy of the virus, including endosomal escape and nuclear localization. For clinical usage of AAVs the abundance of VP1 and VP2 can be optimized to gain efficacy, but then methods to assay the virus composition are essential. The high-resolution native mass spectrometry data of AAVs presented here, together with the simulations thereof based on the stochastic assembly model, provide one of the most accurate means to determine these stoichiometries and VP distributions, and can therefore become an important tool for quality control of AAV vectors.

## Methods

**AAV capsid preparation**. VP3-only AAV1, wild-type AAV1 and AAV8_1 were produced and characterized via SDS-PAGE and negative stain EM[48]. Briefly, fractions of samples produced using a stable baculovirus/SF9 cell line and purified using an AVB sepharose column (GE Healthcare) were concentrated and loaded onto a 5–40% step sucrose gradient to separate empty and full (DNA containing) capsids. The capsids were separated by centrifugation at 151,000xg (at $r_{average}$ in an SW41 rotor) for 3 h at 4 °C. The empty capsids were extracted from the 20-25% sucrose fraction and the full capsids from the 30-35% fraction. The samples were buffer exchanged into 1XTD buffer and concentrated in an Apollo concentrator (Orbital Biosciences) and the sample concentrations determined by UV spectrometry for the empty capsids ($E = 1.7$ for concentration in mg/ml). The purity and capsid integrity were confirmed by SDS-PAGE and negative stain Electron Microscopy (EM), respectively. AAV5 and AAV8_2 capsids were purchased from Virovek (Hayward, CA, USA), produced in SF9 cells following their patented BAC-to-AAV technology. AAV9 samples were produced in HEK293 cells using a triple-transfection approach.

**RP-UHPLC/MS of VP monomers**. AAV5, AAV8_2 and AAV9 were subjected to reversed-phase ultrahigh-performance liquid chromatography/ ultrahigh-resolution electrospray ionization quadrupole time-of-flight mass spectrometry (RP-HPLC-UV/MS) for quantitation of VP capsid protein ratios and VP protein characterization at the intact level. Samples were reduced with a 1:1 (v/v) mixture of TCEP, denatured on column and chromatographically separated on a Waters Acquity BEH C4, 1.7 μm, 2.1x100mm, 300 Å narrowbore column using a Waters HClass UHPLC. The column was held at a temperature of 50 °C with a gradient consisting of 20%-90% organic mobile phase over 75 min with a flow rate of 0.2 mL/min. The aqueous mobile phase consisted of 0.1% trifluoroacetic acid (TFA) in water and the organic mixture consisted of 50% 2-propanol 50% acetonitrile with 0.08% trifluoroacetic acid. The Waters HClass UHPLC was coupled to a Bruker Daltonics maXis II electrospray ionization quadrupole time-of-flight mass spectrometer.

VP ratio quantitation was performed using both UV 214 nm and MS data. Initially, peaks in the UV chromatogram that were related to AAV capsids proteins were integrated in Bruker Data Analysis to compute a relative peak area for each component. Due to insufficient chromatographic separation required to fully resolve capsid proteins, the contribution of individuals VP proteins within a given UV peak was computed with the deconvolved MS signal intensity.

**Native MS**. Prior to native MS analysis, samples were buffer exchanged to aqueous ammonium acetate (75 mM, pH 7.5) with several concentration/dilution rounds using Vivaspin Centrifugal concentrators (50 kDa MWCO, 9,000 $g$, 4 °C). An aliquot of 1–2 μl was loaded into gold-coated borosilicate capillaries 467 (prepared in-house) for nano-ESI. Samples were analyzed on a standard commercial Q Exactive-UHMR instrument (MS Tune QE-UHMR 2.11, Thermo Fisher Scientific)[37,49]. Instrument parameters were optimized for the transmission of high mass ions. Therefore, ion transfer target $m/z$ and detector optimization were set to 'high $m/z$'. In-source trapping was enabled with a desolvation voltage of −50 V and the ion transfer optics (injection flatapole, inter-flatapole lens, bent flatapole and transfer multipole) were set to 10, 10, 4, and 4 V, respectively. Xenon was used as collision gas for all experiments in the range of $8\times10^{-10}$ to $2\times10^{-9}$ mbar (UHV readout). Particles were desolvated in the HCD cell with HCD energies ranging between 100 and 130 V. Data was acquired at resolution settings corresponding to 32 and 128 ms transients with transient averaging enabled.

**Native MS spectra simulations**. For the simulation of native mass spectra a python class was developed capable of creating theoretical mass spectra of complex samples while considering the inverse square root dependency between $m/z$-positions and resolution as present in the Orbitrap mass analyzer. For the simulation of charge state series originating from a single mass (as shown in Fig. 2b, c) the $m/z$-position for each ion is calculated for a defined charge state range and the relative intensities are calculated assuming a Gaussian charging distribution. Next, an empty intensity array is created where the index corresponds to the simulated $m/z$-range at a defined data point density. For each ion, we calculated the theoretical peak shape considering the theoretical resolution at its given $m/z$-position.

This is done by using the Gaussian probability density function with $\mu = m/z$-position and $\sigma = FWHM(m/z$-position$) / 2.355$. The probabilities/intensities are calculated for the corresponding $m/z$-bins within 3 times the FWHM of the centroids $m/z$-position and then added to the intensity array at its corresponding position. This step is repeated for each ion species in the charge state distribution. After all ion species are added to the intensity array we performed a baseline correction step in order to mimic transient averaging as applied to the experimentally recorded spectra. If not stated differently, all showed simulations in these studies were subjected to baseline correction.

For more complex spectra, containing ions species from more than one mass, the first step is the calculation of all containing masses and their relative abundances as displayed in Fig. 3b. The average charge for each mass was calculated following the empirical determined formula[50] $z = 1.638 \times MW^{0.5497} + b$, with $b$ being an offset to align simulated and experimental average $m/z$-position. The width was held constant for all calculated charge state distributions. From there, the Gaussian peaks are calculated for each charge state distribution as described above for individual masses. The intensities for each charge state distribution are scaled according to the relative abundance of its corresponding mass and then added to the final mass spectra. See Supplementary Fig 2 for illustration of the simulation procedure.

**Simulation screening**. Simulation screenings were carried out by changing the bulk expression levels of VP1 and VP2 by increments of 1 from 0 to 100% (VP3 percentage is defined by VP3% = 100%-VP1%-VP2%). For each bulk expression level we calculated the mass distribution using the multinomial model for all 1891 combinations as depicted in Fig. 3. Since the average mass of the resulting mass distributions can differ by more than one megadalton, as in the case of 100% VP1 or VP3 expression (4.7 MDa vs 3.5 MDa), the offset constant for the charging $b$ has to be adjusted for each simulation so the simulated charge state distribution populates the same $m/z$-region as the experimental spectra. The offset constant is calculated by dividing the average mass by the average $m/z$-position the experimental spectrum populates and subtract it by the charge obtained for the average mass by using the formula $z = 1.638 \times MW^{0.5497}$. We added to all final capsid masses +2.8 kDa to account for solvent adducts as observed in the VP3 only capsid shown in Fig. 2b. See Supplementary Fig 2 for illustration of the simulation procedure.

**Simulation scoring**. After calculating the mass spectra for various VP expression levels we compared them with the experimentally obtained data. We found that peak heights are not a reliable scoring parameter as it is influenced by transient averaging, baseline correction, and the estimated charging offset used in the simulations. Therefore, peak centroids were extracted from the experimental spectrum and peaks were matched with the closest centroid in the simulation, within half the average peak distance. The average deviation was calculated for all matched peaks and we penalized all peaks which were not matched in the experimental data or the simulation by adding 1 Th to the average deviation. Average peak deviations were converted to scores using a Gaussian probability density function ($\mu=0$, $\sigma=10$) normalized to return 1 for 0 Th average peak deviation. See Supplementary Fig 2 for illustration of the simulation procedure.

**Reporting summary**. Further information on research design is available in the Nature Research Reporting Summary linked to this article.

## Data availability

Data supporting the findings of this manuscript are available from the corresponding author upon reasonable request. A reporting summary for this Article is available as a Supplementary Information file. Source data are provided with this paper.

## Code availability

A python script for the simulation and scoring of complex AAV mass spectra is available as Supplementary Software.

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

## Acknowledgements

We thank the members of the Heck laboratory for general support, especially Arjan Barendregt. This research received funding through the Netherlands Organization for Scientific Research (NWO) TTW project 15575 (Structural analysis and position-resolved imaging of macromolecular structures using novel mass spectrometry–based approaches) and the Spinoza Award SPI.2017.028 to A.J.R.H. A.B. and M.A.-M. are supported by NIH R01 GM109524, NSF DMS 1563234, and funds from the UF College of Medicine. J.S. is supported by the Dutch Research Council NWO Gravitation 2013 BOO, Institute for Chemical Immunology (ICI, 024.002.009). Additionally, we are grateful for the support from the Pfizer Biotherapeutics Pharmaceutical Sciences organization.

## Author contributions

T.P.W., J.S., and A.J.R.H. conceived the project, designed the experiments and wrote the paper. S.H. performed initial native MS experiments. O.F., T.P., A.B., and M.A.-M. provided the AAV samples. A.B and M.A.-M. performed negative stain EM and SDS-PAGE based characterization. O.F. and T.P. performed RP-UHPLC/MS of the VP monomers. T.P.W. wrote the scripts for spectra simulation and analyzed all the mass spectrometry data. J.S. and A.J.R.H. supervised the project. All authors discussed the results and edited the paper.

## Competing interests

O.F. and T.P. are employees of Pfizer WRDM, St Louis, MO, a company with interest in employing AAV vectors for gene delivery purposes. M.A.-M. is co-founder of StrideBio, Inc., with an interest in developing AAV technology for gene delivery purposes. M.A.-M. is a member of the ATGC and Voyager SAB, and consultants for StrideBio, Intima Bioscience, being biopharma companies with interest in developing AAV for gene delivery purposes. The remaining authors declare no competing interests.
