## [Peer Review File · Nature Communications]

Reviewer #1 (Remarks to the Author):

Recombinant adeno-associated virus (rAAV) gene delivery vectors have become central to developments in human gene therapy. rAAV capsids consist of three viral proteins (VP1, VP2, and VP3) with known total stoichiometry, but it is unclear whether recombinant AAV capsids have a uniform composition or whether they are a heterogeneous mixture of capsids with differing stoichiometry. The authors have approached this problem through native mass spectrometry (MS) experiments and spectrum simulation, and came to the conclusion that rAAV capsids represent a heterogeneous mixture of stochastic combinations of viral proteins. This finding is of great importance to human gene therapy research, and the spectrum simulation approach pushes native MS beyond current limits of mass resolution. Specific comments are listed below.

- for readers who are less familiar with rAAV research, an upfront summary of the exact function of each of the three VPs would be helpful, and a more compact discussion of any possible effects of capsid heterogeneity on GOI delivery in the introduction would clarify the motivation for the study and increase the impact of the paper.

- references should be added, e.g., for the sentence “These N-terminal extensions of VP1 and VP2 are reported to play crucial roles in endosomal trafficking and escape, nuclear localization, and genome release” (page 4)

- page 3: it would be clearer to say “AAV capsids consist of a total of 60 molecules of viral proteins” instead of “AAV capsids consist of a total of 60 viral proteins”

- could the +2.8 kDa mass difference result from something other than solvent, i.e., is it possible that the capsids contain species other than VPs? Was this, or a similar mass difference, observed in other native MS studies of MDa particles?

- in reference 27 (J. Am. Chem. Soc. 2014, 136, 7295-7299), a comparison of simulated and experimental spectra of AAV1 led to the proposal that capsid assembly is stochastic such that VP1/VP2/VP3 stoichiometry depends mainly on the relative expression levels of VPs. It would be helpful to more clearly point out the difference between the spectrum simulations in reference 27 and this manuscript. Figure 2 in this manuscript (which is very similar to Figure 4 in reference 27) shows simulated spectra for different AAV1 stoichiometry, but what does the ‘stochastic’ simulated m/z-spectrum look like? Difference spectra (experimental minus simulated) for both the ‘stochastic’ simulated spectrum and a spectrum that simulates the interpretation of the earlier report (reference 27) should be included to demonstrate that the stochastic model shows better agreement with experiment than other models. Statistics of the difference spectra (average values and standard deviations) will then provide a quantitative measure for the agreement between different simulated spectra and the experimental data. What do the authors mean by “somewhat ambiguous” (page 8) when referring to reference 27?

- the simulated spectra in Figure 4b show unresolved humps (which were removed by baseline correction), but why were these humps not observed in the experimental spectra? How does transient averaging remove humps? Signals should not disappear because they are unresolved. If they did, how can any of the spectra be interpreted in terms of relative abundance and capsid stoichiometry?

- could single-particle MS experiments confirm the heterogeneity of rAAV capsids?
- would it be possible to incrementally move a small isolation window through the m/z range, to dissociate capsids at each position, and to detect the corresponding VPs for reconstruction of capsid stoichiometry?
- page 10/11: I had trouble understanding the sentence "The simulated spectrum for the stochastic assembly model, using the bulk VP1:VP2:VP3 ratio of 3:10:47 (determined from LC-UV/MS data) closely resembles our experimental data, supporting that it is indeed an accurate description of the AAV capsid assembly" - did the bulk ratio go into the scoring simulation? If yes, what is the meaning of Table 1?
- what is the reason that higher-resolution spectra (128 ms transient time) could be obtained for AAV9 but not the other capsids?

Reviewer #2 (Remarks to the Author):

The manuscript by Wörner, et al., describes a novel application of mass spectrometry to demonstrate that in a normal (i.e. not baculovirus expressed capsid protein) adeno-associated viral infection, where there are three versions of the capsid protein, the capsid proteins are stochastically chosen to assemble into the final particle.

The manuscript is well written and the findings are sound. However, I struggled with whether the biological impact of the findings would be appropriate and of sufficient general interest to the Nature reader. The paper does a good job in presenting a novel method to look at entire viral capsid distributions and is worthy of publication, but in a more specialized journal where the results can be better appreciated.

REVIEWER COMMENTS

Reviewer #1 (Remarks to the Author):

Recombinant adeno-associated virus (rAAV) gene delivery vectors have become central to developments in human gene therapy. rAAV capsids consist of three viral proteins (VP1, VP2, and VP3) with known total stoichiometry, but it is unclear whether recombinant AAV capsids have a uniform composition or whether they are a heterogeneous mixture of capsids with differing stoichiometry. The authors have approached this problem through native mass spectrometry (MS) experiments and spectrum simulation and came to the conclusion that rAAV capsids represent a heterogeneous mixture of stochastic combinations of viral proteins. This finding is of great importance to human gene therapy research, and the spectrum simulation approach pushes native MS beyond current limits of mass resolution.

We appreciate these positive comments by this expert reviewer very much

Specific comments are listed below.

- for readers who are less familiar with rAAV research, an upfront summary of the exact function of each of the three VPs would be helpful, and a more compact discussion of any possible effects of capsid heterogeneity on GOI delivery in the introduction would clarify the motivation for the study and increase the impact of the paper.

Thank you very much for the comment. We changed the corresponding section in the manuscript (see line 128-134).

- references should be added, e.g., for the sentence “These N-terminal extensions of VP1 and VP2 are reported to play crucial roles in endosomal trafficking and escape, nuclear localization, and genome release” (page 4)

We included a reference (see line 76).

- page 3: it would be clearer to say “AAV capsids consist of a total of 60 molecules of viral proteins” instead of “AAV capsids consist of a total of 60 viral proteins”

Done (see line 63).

- could the +2.8 kDa mass difference result from something other than solvent, i.e., is it possible that the capsids contain species other than VPs? Was this, or a similar mass difference, observed in other native MS studies of MDa particles?

Indeed, similar mass differences are observed in other native MS studies. We always observe an excess mass of several kDa due to poor desolvation in MDa particles. Below are some recent examples from previously published work:

1. FHV 9.4 MDa -> 18 kDa solvent adducts (1)
2. AaLS-neg 3 MDa -> 9 kDa solvent adducts (2)
3. BMV (with RNA2 and RNA3+4) 4.5 MDa -> 3-5 kDa solvent adducts (3)

This shows that proper desolvation is an issue for all these complexes and gets generally more challenging with higher mass samples. The here reported 2.8 kDa is actually in the lower range when comparing it with these other particles in the MDa range. Thus, we argue that it is indeed due to incomplete desolvation. Considering this expected additional mass due to incomplete desolvation, any capsid component beyond the three VPs is sure to be very minor in its mass contribution or to be present in mere trace amounts.

- in reference 27 (J. Am. Chem. Soc. 2014, 136, 7295-7299), a comparison of simulated and experimental spectra of AAV1 led to the proposal that capsid assembly is stochastic such that VP1/VP2/VP3 stoichiometry depends mainly on the relative expression levels of VPs. It would be helpful to more clearly point out the difference between the spectrum simulations in reference 27 and this manuscript. Figure 2 in this manuscript (which is very similar to Figure 4 in reference 27) shows simulated spectra for different AAV1 stoichiometry, but what does the 'stochastic' simulated m/z-spectrum look like? Difference spectra (experimental minus simulated) for both the 'stochastic' simulated spectrum and a spectrum that simulates the interpretation of the earlier report (reference 27) should be included to demonstrate that the stochastic model shows better agreement with experiment than other models. Statistics of the difference spectra (average values and standard deviations) will then provide a quantitative measure for the agreement between different simulated spectra and the experimental data. What do the authors mean by "somewhat ambiguous" (page 8) when referring to reference 27?

The reviewer is right. Figure 4 in reference 27 and Figure 2 in this manuscript are indeed quite alike. The analysis of Figure 2 in the current work does relate to a novel dataset of AAV1 acquired on the newer UHMR Orbitrap platform and includes the additional AAV1 VP3-only capsids as an insightful reference sample. Both here and in our previous study we aim to demonstrate the complexity of AAV mass spectra with these figures. This small subset of simulated spectra highlights the features of AAV composition that ultimately determine the fine structure of the spectra. It represented a necessary first insight into MS analysis of intact AAV capsids in our previous work and serves as an illustrative starting point for the more comprehensive and elaborate stochastic assembly model we develop in the current work.

To illustrate the difference in the approach of ref 27 and our current study, please consider the following example. If we take AAV1 and we assign charges to the three peak series we get masses which correspond to a VP stoichiometry of 2:2:56, 2:3:55 and 2:4:54. However, AAV populates a m/z region where a VP3-to-VP1 substitution will produce a mass spectrum which will align almost perfectly over a wide range at this mass resolution. Therefore, also the stoichiometries 3:2:55, 3:3:54 and 3:4:53

could be contributing to these spectra. However, even these 6 stoichiometries are only a small fraction of the 1891 possible stoichiometries. The simulations below illustrate how if we produce a mass spectrum only consisting of these 3 and 6 stoichiometries we obtain a simulation that to some extent resembles the experimental spectra but the peak positions are off and the spacing doesn't match, especially in the high mass range (at m/z 22,500-23,000). However, when we use a much broader multinomial distribution to calculate the probability for all possible stoichiometries we will get a mass distribution, which when simulated as ESI mass spectrum, matches the peak positions of the experimental spectra much better (see below simulation at bottom). Here it should be noted that the only input parameters for calculating the probability for each VP stoichiometry are the bulk VP expression ratios.

In summary, different from the earlier published work (ref 27) where we only used a few possible simulated stoichiometries to explain the fine structure of the spectrum, we now developed a mechanistic model of stochastic AAV capsid assembly, which produces a much-improved qualitative fit to the experimental spectra by allowing all possible stoichiometries to contribute.

Rebuttal Figure 1: Comparison between (from top to bottom) the experimental AAV1 spectrum, a simulation using only 3 stoichiometries based on direct assignments, a simulation with 6 stoichiometries based on direct assignment and potential overlap due to VP3 toVP1 substitutions, and a simulation of all possible 1891

stoichiometries with abundances following a multinomial distribution. Dotted lines indicate regions where the extended (novel) model provides a much better match than the heterogeneity earlier described in ref 27.

- the simulated spectra in Figure 4b show unresolved humps (which were removed by baseline correction), but why were these humps not observed in the experimental spectra? How does transient averaging remove humps? Signals should not disappear because they are unresolved. If they did, how can any of the spectra be interpreted in terms of relative abundance and capsid stoichiometry?

This is indeed a very good question. This “removing humps” effect is caused by transient averaging as well as by the data processing during the eFT. The main difference between transient averaging and spectra averaging is that for the later, noise and signal has only positive values so if they are summed up it will increase the baseline. This is not the case when the transients are averaged, as then the noise will cancel itself out and produce a better S/N in the eFT spectrum, with no elevated baseline. Additionally, if an eFT spectrum shows a stable baseline it is removed from the final spectra during the signal processing workflow. See rebuttal Figure 2.

Rebuttal Figure 2: AAV1 spectra recorded by (left) spectrum and (right) transient averaging.

In general, ion intensities of poorly resolved peaks should not be used quantitatively. For our simulation scoring we were very careful to avoid such signal processing effects; we didn't consider the centroids intensity but rather *only* its m/z position, which is dependent on the relative abundances of the different enveloped ions species. We included the baseline correction step in our spectrum simulations to better represent the experimental spectra, but by ignoring peak intensity altogether we achieved a more robust scoring that is insensitive to these abovementioned processing artefacts.

- could single-particle MS experiments confirm the heterogeneity of rAAV capsids?

A direct confirmation is currently not possible due to the limited mass resolving power. Examining the potential mass distribution for AAV9 in fig 3 we see that we would need a very high resolution to resolve mass differences of less than 3 kDa (red bar). This would require a resolving power of more than 1000. Where the current maximum achieved mass resolving power is roughly 300 at the max (on a dedicated CDMS platform) (4).

Rebuttal Figure 3: (left) Mass histogram for AAV9 following a completely stochastic assembly. (right) A zoom into the fine structure of the distributions illustrating the required mass resolution to resolve individual stoichiometries by using CDMS.

- would it be possible to incrementally move a small isolation window through the m/z range, to dissociate capsids at each position, and to detect the corresponding VPs for reconstruction of capsid stoichiometry?

Such an experiment would be indeed interesting and theoretically possible but very hard to execute and analyze. First, a narrow mass-window selection of such high

mass samples is non-trivial since the quadrupole is located before the HCD cell. At that stage the ions still harbor much more solvent adducts and have to be selected “blindly” since the m/z is higher than it appears in the measured m/z after desolvation. Second, each peak triplet for $t=32$ ms (as shown in Manuscript Fig 4d) consist already of around ~1000 distinct ion species, that thus would be co-isolated. Since asymmetric charge partitioning is stochastic to the precise number of charges the ejected protein will strip the complex of, it will actually increase the heterogeneity further. Additionally, since we select only a fraction of the total ions and spread them over a much wider m/z -range, the S/N will drop significantly too.

- page 10/11: I had trouble understanding the sentence “The simulated spectrum for the stochastic assembly model, using the bulk VP1:VP2:VP3 ratio of 3:10:47 (determined from LC-UV/MS data) closely resembles our experimental data, supporting that it is indeed an accurate description of the AAV capsid assembly” - did the bulk ratio go into the scoring simulation? If yes, what is the meaning of Table 1?

Table 1 is an overview of the analyzed AAV samples and compares the bulk ratio, measured by LC-UV with the bulk ratio which gave the best simulated spectral match. The sentence means that if we simulate a mass distribution from such a bulk ratio, using all possible 1891 stoichiometries (instead of only 3 or 6 see also rebuttal Figure 1) at an abundance defined by the bulk VP1:VP2:VP3 ratio, we get already a mass spectrum which matches the experimental data fairly well in its general appearance (peak series). The systematic comparison of simulations for different bulk VP ratios gives a more refined match (matching the actual peak centroids) and allows to estimate the bulk ratio purely based on the native MS spectra. To make this statement more clear we rewrote the corresponding section (see line 285-290)

- what is the reason that higher-resolution spectra (128 ms transient time) could be obtained for AAV9 but not the other capsids?

Whether an AAV prep can be mass resolved at higher resolutions depends on how the ion signals of all co-occurring particles are distributed and align with each other. Ultimately, this is dependent on the masses of the different VP subunits, the bulk average ratio of the VPs and which m/z region these ions cover in the mass spectra. To illustrate this further, below are depicted the simulated spectra at $t=32$ and $t=128$ for AAV9 and AAV1. For AAV9, we see that we can resolve additional peaks when moving from 32 to 128 ms transients. In contrast, this is not the case for AAV1 when simulated at the m/z position where we detect most of the ions. When moving to another m/z position, we could theoretically resolve additional peaks at higher mass resolution but as we don't have fine control over the charging in nESI this is purely theoretical.

Rebuttal Figure 4: Overlay of simulations for (top) AAV9 and (middle and bottom) AAV1 for 32 ms and 128 ms transients at different m/z -positions. The comparison shows the potential of resolving additional peak series at higher transient times. While the ions signals for the 1891 stoichiometries align for AAV9 in such manner that we can resolve additional peak series at higher transient times.

This is not the case for AAV1 at the experimentally observed m/z region center at 22000 m/z but would be theoretically possible at 24000 m/z.

Reviewer #2 (Remarks to the Author):

The manuscript by Wörner, et al., describes a novel application of mass spectrometry to demonstrate that in a normal (i.e. not baculovirus expressed capsid protein) adeno-associated viral infection, where there are three versions of the capsid protein, the capsid proteins are stochastically chosen to assemble into the final particle.

The manuscript is well written and the findings are sound. However, I struggled with whether the biological impact of the findings would be appropriate and of sufficient general interest to the Nature reader. The paper does a good job in presenting a novel method to look at entire viral capsid distributions and is worthy of publication, but in a more specialized journal where the results can be better appreciated.

We respectively disagree with this reviewer, and appreciate to hear that the first reviewer and editor do also support our view that this work is a good contribution to Nature Communications, in particular as AAV has become such an important vehicle for gene-delivery.

1. van de Waterbeemd M, Fort KL, Boll D, Reinhardt-Szyba M, Routh A, et al. 2017. High-fidelity mass analysis unveils heterogeneity in intact ribosomal particles. *Nat. Methods*, pp. 1–7
2. Fort KL, van de Waterbeemd M, Boll D, Reinhardt-Szyba M, Belov ME, et al. 2017. Expanding the structural analysis capabilities on an Orbitrap-based mass spectrometer for large macromolecular complexes. *Analyst*. 143:
3. van de Waterbeemd M, Snijder J, Tsvetkova IB, Dragnea BG, Cornelissen JJ, Heck AJR. 2016. Examining the Heterogeneous Genome Content of Multipartite Viruses BMV and CCMV by Native Mass Spectrometry. *J. Am. Soc. Mass Spectrom.* 27(6):1000–1009
4. Todd AR, Barnes LF, Young K, Zlotnick A, Jarrold MF. 2020. Higher Resolution Charge Detection Mass Spectrometry. *Anal. Chem.* 15:29

Reviewer #1 (Remarks to the Author):

All points raised in my previous review have been satisfactorily addressed, along with corresponding revisions in the manuscript. The only further suggestion I have is to increase the size of the “Scoring” Figure in the supporting material and to perhaps also include the related Rebuttal Figure 1.

REVIEWER COMMENTS 18.01.21

Reviewer #1 (Remarks to the Author):

All points raised in my previous review have been satisfactorily addressed, along with corresponding revisions in the manuscript. The only further suggestion I have is to increase the size of the "Scoring" Figure in the supporting material and to perhaps also include the related Rebuttal Figure 1.

We appreciate these comments and changed the supporting material accordingly including now the earlier Rebuttal Figure 1.